# Effect of Process Parameters on Thermal and Mechanical Properties of Filament Wound Polymer-Based Composite Pipes

**DOI:** 10.3390/polym15132829

**Published:** 2023-06-27

**Authors:** Sara Srebrenkoska, Filip Kochoski, Vineta Srebrenkoska, Svetlana Risteska, Renata Kotynia

**Affiliations:** 1Faculty of Mechanical Engineering, Goce Delcev University, Krste Misirkov 10-A, P.O. Box 201, 2000 Stip, North Macedonia; sara.srebrenkoska@ugd.edu.mk; 2Faculty of Technology, Goce Delcev University, Krste Misirkov 10-A, P.O. Box 201, 2000 Stip, North Macedonia; filip.kocho@gmail.com; 3Institute for Advanced Composites and Robotics (IACR), 7500 Prilep, North Macedonia; svetlanar@iacr.edu.mk; 4Faculty of Civil Engineering, Architecture and Environmental Engineering, Lodz University of Technlogy (TUL), 93-590 Lodz, Poland; renata.kotynia@p.lodz.pl

**Keywords:** polymer composite, filament winding, experimental design, thermal analysis, mechanical properties

## Abstract

The aim of this study was to investigate the mechanical and thermal properties of composite pipes based on epoxy resin and glass fibers produced by filament winding (FW) technology. Epoxy resins are widely used polymers in FW composite structures. The thermal characterization of the neat epoxy resin, curing, and post-curing characteristics for the determination of polymerization and glass transition temperature was performed, which is important for the mechanical properties of polymer composite pipes. In the present work, the applicability of the full factorial experimental design in predicting the hoop tensile and compressive strengths of glass fiber/epoxy resin composite pipes was investigated. The composite pipes in accordance with the 2^3^ full factorial experimental design by using of three parameters and two levels of variation were prepared. The winding speed of the composites was taken to be the first factor, the second was the fiber tension, and the third was winding angle. To approximate the response, i.e., the mechanical properties of the composite pipes within the study domain, the first-order linear model with the interaction was used. The influence of each individual factor to the response function was established, as well as the influence of the interaction of the two and three factors. Additionally, those results were completed with the thermal characterization of the polymer composite pipes. From received results from mechanical and thermal characterization, it was concluded that the properties of composite specimens were highly affected by the analyzed parameters in filament winding technology. It was found that the estimated first-degree regression equation with the interaction gave a very good approximation of the experimental results of the hoop tensile and the compressive strengths of composite pipes within the study domain.

## 1. Introduction

Filament winding (FW) technology is one of the manufacturing processes with a high degree of excellence and automation that has applied to production of composite structures for gas storage and transportation [1,2,3]. Filament winding is an effective fabrication technique for creating cylindrical composite structures such as tubing and chemical and fuel storage containers [4,5,6]. Various pressure vessels have evolved in the last few decades, from metal to fiber-reinforced tanks, primarily for weight saving and high pressure resistance. Composite pressure vessels (CPVs) can affect fuel gas tank weight savings up to 75% compared to metallic vessels [6,7]. As a result, manufacturing of composite pipes and CPV through FW technology has been more frequently applied [2,8,9].

In the filament winding method, fiber strands are unwound and passed continuously to the resin tank. In the resin tank, fiber strands are impregnated completely with the resin. Then, these resin-impregnated strands are passed onto a rotating mandrel. These strands are wound around the mandrel in a controlled manner and in a specific fiber orientation [10,11]. Fiber tension is critical in filament winding because compaction is achieved through the fiber tension. The fiber tension affects the percentage of fiber reinforcement and porosity content in the composite, which in turn affects the properties of the processed composite product. The fiber tension depends upon the type of fiber, its geometry, and the winding pattern required on the rotating mandrel [12,13,14]. The fiber tension should be at an optimal level because too-high fiber tension may break the fiber completely or initiate fiber fracture at the surface. Curing of the composite is done with heat, generally in an oven, and the final composite product is taken out of the mandrel [15,16]. To remove the metallic mandrel from the composite part, hydraulic rams may be used. A high-fiber volume fraction can be achieved in the composite with this processing technique. Nowadays, computer-controlled machines are used which independently monitor every movement of the whole process [17,18].

The quality of filament-wound products can be affected by various parameters, so the choice and combination of these factors are essential for minimizing output difficulties and increasing structural performance [12,13,14]. Besides material engineering properties, fiber direction, fiber tension, and winding velocity are necessary to control when designing filament-wound composites [10,11]. In filament winding, one can vary winding tension, winding angle, and/or resin content in each layer of reinforcement until the desired thickness and strength of the composite are achieved. Though many design and manufacturing challenges are associated with various process factors involved in winding technology, careful considerations are needed to create a reliable product. Therefore, it is essential to comprehend the various process parameters, their combined effects, and the associated challenges while designing and fabricating filament-wound structures [19,20,21].

Some combinations of optimization techniques such as design of experiment (DOE) have been used for optimizing the design of composite structures in terms of specific strength, failure behavior, geometrical parameters, and hence for reducing the overall cost of the composite structure. In the last few years, “Taguchi methods” and “Full factorial experimental design” are used frequently in industry, especially in various stages of product development and in process and quality improvement, and we have incorporated some of these ideas [19,20,21]. We have implemented full factorial experimental design (FFED) for analyzing the variability of the response (in our case, hoop tensile strength and compressive strength) as the factors vary in some intervals of variation. It has the advantage of saving time and money in running the experiment. The factorial experimental design may be run for one or more of the following reasons: to determine the principal causes of variation in a measured response, to find the conditions that give rise to a maximum or minimum response, to compare the responses achieved at different settings of controllable variables, and to obtain a mathematical model in order to predict future responses. The full factorial experimental design allows mathematical modeling of the investigated process in a study domain in the vicinity of a chosen experimental point [22].

This paper presents the effect of some process parameters (winding speed, tension of the fibers, and winding angle) on the mechanical behavior of filament-wound glass/epoxy composite pipes. In addition, a full factorial experimental design with statistical analysis strategies is summarized with related disputes and suggestions. The selected factors are the most influential factors and their levels of variation were chosen based on industry experience, the design of the machine used for this technology, and of the real problems that occur during production of the composite pipes. The rotating speed of the mandrel determines the dwelling time of the filament strand in the resin bath so that all the filaments are thoroughly wetted [23]. On the other hand, it also enables a precise layup of the impregnated strand on the rotating mandrel, i.e., the previous layer without any gaps or overlapping [24,25]. Due to the deposition of multilayers, there is a cyclic profile of temperature in the FW process, which is considerably related to fabrication and, consequently, with the curing kinetics of epoxy resin systems, curing temperature, and strength of the manufactured parts [26,27,28,29,30,31,32,33,34]. In parallel with the study of the process parameters effect, a few thermal analyses (TGA and DSC) of epoxy resin and composite specimens have been measured to determine the range of curing temperature, degradation of the composites, glass fiber weight percentage, and volatile materials, i.e., thermal and dimensional stability of composite pipes. The potential application of these composite pipes is for transportation of hot fluid under pressure. That is the reason for determining the mechanical as well as thermal properties of these pipes.

The preliminary results related to thermal and mechanical properties revealed that differences in crystallinity percentage exist and stress/strain failure can be considered as an indicator to evaluate the mechanical properties of FW manufactured pipes. The results confirm the impact of the mentioned parameters’ roles on the bonding formation in the FW process. Therefore, the selection of the optimized and suitable process parameters is an important design consideration.

## 2. Experimental Part

### 2.1. Materials

In this study for the production of the composite pipes, 10 bobbins of E-glass fiber roving 185P from Owens Corning, Brussel, Belgium were used. The glass fibers were impregnated into an Araldite LY1135/Aradur 917/Accelerator 960 epoxy resin system from Huntsman Interlational LLC, Houston, TX, USA. Samples with different winding parameters were wound on an iron mandrel with pins on both sides with help of a MAW FB 6/1 laboratory filament winding machine with six axes and a roller type resin bath manufactured by Mikrosam D.O.O, Prilep, North Macedonia. The materials’ main parameters are shown in Table 1 and Table 2.

### 2.2. Preparation of Composite Pipes and DOE

The wet winding method was adopted. During the impregnation of the fibers, several factors were observed (resin viscosity, speed of impregnation, temperature) so that the required resin content in the composite pipes was attained. The preparation of the composite pipes was conducted in accordance with the 2^3^ full factorial experimental design (FFED) by using three parameters and two levels of variation. In the FFED, the winding speed of the glass fibers was taken to be the first factor, the second factor was fiber tension, and the third factor was winding angle. For the first factor, the low and high levels were set at 5.25 m/min and 21 m/min, respectively, for the second factor at 34 N and 60 N, respectively, and for the third factor at 10° and 90°. The values of each factor have been chosen as the lowest and highest values which can be applied to the production procedure. Test values were the extreme values which were achievable on the filament winding machine that we were using to produce the samples. With that assumption, we have taken the first-order linear model with interactions to predict the response function, i.e., the hoop tensile and compression strengths of the composite pipes within the stated study domain (5.25–21) m/min × (34–60) N × (10–90)°. In accordance with the FFED procedure, 8 (2^3^) trials are needed, i.e., all possible combinations of the variables were tested. The value 2^3^ has the following meaning: the exponent 3 shows the number of factors, while the base 2 shows the number of levels that each factor takes. Therefore, the total number of experiments is 8, with 5 replications for each trial and a total of 40 trials, i.e., for each combination of the trial factors, 5 trials are made. The coding of the variables is conducted in accordance with Table 3.

The FFED allows mathematical modeling of the investigated process in the vicinity of a chosen experimental point within the study domain [1]. That is the center point (a zero level, *x_i_ =* 0) in the whole range of each variable, and if we add/subtract the interval of variation to this point (zero level), we will get the higher and the lower level, respectively, for that variable. For the winding speed of the composite pipes, we have chosen the central point as 13.125 m/min, 47 N for the fiber tension, and the central point 50° (which corresponds to previously defined levels) for the winding angle. The winding angle is the angle between the fiber and the line on the surface of the mandrel, which is parallel to the mandrel axis. The coupled helical winding of layers (±θ) is usually preferred, whereas a hooped winding angle, very close to 90° and winding with very low winding angle, can also be used. By varying the winding angle with respect to the mandrel axis, directional strength can be obtained by considering the loads, which will operate on the finished product [2,6]. The calculated mass ratio between fiber and resin of the produced composite pipes was 75:25%wt. After the winding process, the samples were cured at 80 °C and at 140 °C for four hours. After the curing, the mandrel was removed from the specimens. The pipes and NOL ring specimens’ preparation processes are shown in Figure 1 and Figure 2.

### 2.3. Thermal Analysis

The thermal characteristics of the epoxy resin and of the different models of filament-wound glass fibers/epoxy composites were measured using both differential scanning calorimetry (DSC) and thermogravimetric analysis (TGA) methods. The TGA measurements were performed with a Mettler–Toledo TGA instrument (Figure 3a). About 20 mg of each sample was heated from 50 °C to 1000 °C at a heating rate of 20 K/min under argon/air flow of 50 mL/min. The DSC measurements were performed with a Mettler–Toledo DSC instrument (Figure 3b). The experiments were carried out under a constant flow of nitrogen of 50 mL/min. For the dynamic DSC measurements, the epoxy resin system and the composites were subsequently heated at a constant rate of 5, 10, 15, and 20 °C/min over a temperature range of 20 to 350 °C.

### 2.4. Mechanical Performance Test

An investigation of the mechanical properties of composite pipes was performed as well as the effect of winding parameters on hoop tensile and compressive strengths. The tensile testing was performed according to ASTM D2290 [35] using split-disk test specimens, while the compression testing was conducted according to ASTM D5449 standard [36] using tubular test samples, as shown in Figure 2. The tensile tests were performed at room temperature using a Zwick/Roell universal testing machine with a max load of 50 kN and another one with a max load of 400 kN and loading speed of 0.3 inch/min. The compression tests were also performed at room temperature using the same Zwick/Roell universal testing machine with a max stress of 400 MPa and a loading speed of 0.05 inch/min.

## 3. Results and Discussion

### 3.1. Thermal Analysis

#### 3.1.1. Thermogravimetric Analysis (TGA) 

The results from the thermogravimetric analyses of an uncured epoxy resin are presented in Figure 4. An uncured epoxy resin system loses about 8% of its weight until 350 °C, followed by an ongoing 90% weight loss to 450–500 °C. After that, the weight loss continues with a slower degradation rate. It should be noted that at a temperature of 600 °C, the epoxy resin system exhibits a residual weight of about 8%.

Thermogravimetric (TGA) curves for the composite samples are shown in Figure 5, whereas TGA results are summarized in Table 4. As can be observed, thermal degradation of all composites indicates two stages of the weight loss process. The first stage, occurring in the temperature range from 150 °C to 600 °C, is correlated to the degradation of the epoxy resin and for all composites is about 20% mass. These data are in accordance with the findings for the thermal stability of the epoxy resin. For the second stage, which occurs in the temperature range from 600 °C to 1000 °C, a small shoulder can be noticed which corresponds to the thermal degradation of glass fibers. Since the degradation process occurs in two steps, it can be explained by the degradation phenomena associated with the different composite components.

#### 3.1.2. Differential Scanning Calorimetry (DSC) Analysis

(a) DSC of an uncured epoxy

DSC results of an uncured epoxy resin system are obtained by heating an uncured epoxy resin system at a rate of 5, 10, 15 and 20 °C/min over a temperature range of 20 to 350 °C under a constant flow of nitrogen of 50 mL/min (Figure 6). The onset temperature (Ti), peak temperature (Tp), and termination temperature (Tf) of the curing exothermic peak of the epoxy resin at different heating rates were obtained based on DSC diagrams.

The graphs show the heat flow as a function of the sample temperature. The onset of cure is the temperature at which the heat flow deviates from the linear response and the peak temperature reflects the maximum rate of curing of the resin. At the completion of curing or crosslinking, the DSC heat flow returns to a quasilinear response. The area of the peak can be integrated to give the heat of cure, ∆Cure (J/g). Table 5 shows the specific values of Ti, Tp, and Tf at different heating rates and their extrapolation results.

Based on the DSC scans for an uncured epoxy resin system at different heating rates and from the DSC analysis data shown on Table 4, it can be noticed that two exothermic peaks were observed at heating rates of 5 °C/min and 10 °C/min. In the first peak, the temperature (Tp) is observed at 103.50 °C and 148.2 °C/min, respectively, on the heating rates of 5 and 10 °C/min, whereas in the second peak temperature (Tp) has increased to 118.14 °C and 150.38 °C/min, respectively. However, from DSC analysis data obtained by heating an uncured epoxy resin system at rates of 15 and 20 °C/min, an exothermic peak was observed at 126.12 °C and 132.80 °C, respectively.

As seen in Figure 6/Table 5, the following conclusions can be drawn from the DSC diagrams of the epoxy resin systems. The onset temperature, peak temperature, and termination temperature of the resin system increase as the heating rate rises; a similar analysis and discussion were presented in the research work of Jianguo Liang et al. [33]. With the increase in the heating rate, the curing temperature rises, the curing rate increases, and the curing time is shortened. As seen from Figure 7, according to the least squares method, three temperature values (Ti, Tp, Tf) can be obtained when the heating rate is zero. The extrapolated temperatures are the theoretical start, peak, and end temperatures of the epoxy resin system, which will be used as a reference for resetting the temperature values of the curing conditions.

The kinetic reaction parameters (activation energy Ea, pre-exponential factor A, and reaction model f (α)) are determined using a model-free estimation method designed by Kissinger [29] based on the following equation:
(1)
$$E=-R\frac{d\left(\ln\left(\frac{q}{T_{p}^{2}}\right)\right)}{d\left(T_{p}^{-1}\right)}$$


The apparent activation energy can be obtained by fitting the linear relationship between Kissinger’s 
$\ln(\frac{\beta }{T_{p}^{2}})$
 versus 
$\frac{1}{T_{p}}$
, as shown in Figure 8. The apparent activation energy obtained using Kissinger’s method is an average value of 53.856 kg/mol, which is obtained as the apparent activation energy of the epoxy resin Araldite LY1135/Aradur 917/Accelerator 960.

(b) DSC of a cured epoxy in GFRP

The results of DSC analyses of the filament-wound pipe samples 7-4 and 8-4 are presented on Figure 9 and Figure 10. In addition, DSC results of the analyzed filament-wound pipes are summarized in Table 6.

Based on the DSC scans for different configurations of filament-wound pipes at different heating rate, it can be noticed that the glass transition temperature (Tg) increases with the increasing of the heating rate. The values for Tg of the analyzed composites are similar for all configurations of filament-wound pipes, which indicate that the degree of crosslinking has already been reached in all composites.

As can be observed, thermal degradation of all composites indicates two stages of the weight loss process. The first stage, occurring in the temperature range from 150 °C to 600 °C, is correlated to the degradation of the epoxy resin and for all composites is about 20% mass. These data are in accordance with the finds for the thermal stability of the epoxy resin. For the second stage, which occurs in the temperature range from 600 °C to 1000 °C, a small shoulder can be noticed which corresponds to the thermal degradation of glass fibers. Since the degradation process occurs in two steps, it can be explained by the degradation phenomena associated with the different composite components.

### 3.2. Mechanical Properties and DOE

The width and thickness of each split-disk and tubular specimen were measured with a micrometer (with reading to at least 0.0254 mm). The specimens were tested in the Laboratory of Building Materials and Structural Elements in Lodz University of Technology (TUL) until rupture. The assembled tensile and compression test fixtures and specimens are illustrated on Figure 11.

The apparent hoop tensile strength of the specimens was calculated according to Equation (2):
(2)
$$\sigma =\frac{F_{\max}}{2\cdot A_{m}}$$

where *σ* is the ultimate hoop tensile strength (MPa), *F*_max_ is the maximum load prior to failure (N), and *A_m_* is the minimum cross-sectional area of the two reduced sections, d × b (mm^2^).

The tensile strength of composite rings in each series was determined for specimens according to Equation (2). The overall results for the tensile properties of composite pipes with an experimental matrix are presented in Table 7.

By implementing the 2^3^ full factorial experimental design, we found out that the response function in coded variables, *y_tk_*, is

(3)
$$y_{tk}=419.3+25.2275x_{1}+15.7025x_{2}+398.9138x_{3}+24.4538x_{1}x_{3}$$

and in engineering or natural (real) variables, *y_tn_* is

(4)
$$y_{tn}=-127.2133-0.678x_{1}+1.2079x_{2}+8.9539x_{3}+0.0776x_{1}x_{3}$$


The procedure made the conversion from coded variables into engineering (real) values by showing the real meaning of the regression equation. Then, we used conversion equations for each variable to convert them from coded variables to engineering units (Equation (4)).

In the FFED, the term *x_1_x_3_* is the interaction between factors *x_1_* and *x_3_* which also might have influence on the response, in our case hoop tensile strength (*σ*_value_).

Analyzing the regression equations, it can be found out that the main positive contribution to the hoop tensile strength is given by the winding angle of the of the fibers, i.e., hoop tensile strength is directly proportional to the winding angle of the fibers of the composite pipes. On the other hand, the winding speed of the fibers has a much lower positive effect on hoop tensile strength, but the influence of the fiber tension is the lowest. The interaction of the two factors, for the tensile tests, with a coefficient of +24.4538 also has a negligible positive effect on the response, which is much lower compared to the influence of the winding angle. The test results summarized in Table 7 indicated that the split-disk specimens wound with a 90° angle had the highest values of hoop tensile strength. On the contrary, the specimens of series wound with a 10° angle showed much lower values (only 3% of the strength of series wound with angle 90°). Therefore, the interaction of the two factors *x_1_x_3_*, as well as the influence of the factors *x_1_* and *x_2_* separately, have a negligible influence on the response, i.e., hoop tensile strength. The variation in the burst performance with a changing winding angle configuration is expected, as the composite materials show an anisotropic behavior under different loading conditions. Being highly anisotropic, when loaded in the direction of the fiber alignment, composite materials are likely to exhibit the best mechanical properties. For hoop loading conditions, as the winding angle increases, the fiber alignment direction becomes closer to the loading direction, and for a 90° winding configuration, where the loading direction is the same as the fiber alignment direction, it shall show a maximum. On the other hand, as the winding angle increases, the resistance of the composite tube to axial load decreases. Internal pressure loading is a combined loading type, which creates both axial and hoop stresses in the material, and thus the maximum performance is expected to be attained in the winding configuration, which optimizes the resistances to hoop and axial stresses [8,9]. The increase in the burst performance of the test pipes with an increasing winding angle configuration is thus expected and can be explained by higher hoop resistances being dominant over lower axial resistances as the winding angle configuration increases. After the maximum, the detected decrease can be said to be due to the dominant lowering in axial mechanical resistance over increasing hoop resistance. Considering this, 90° being the best winding configuration is an expected result, as it can be verified from several data in the literature [3,4,5,6,7].

Figure 12 shows a typical tensile force-displacement diagram for samples with the lowest values for the hoop tensile strength (GFRC series 1 and 2), while Figure 13 shows a typical tensile stress–strain diagram at an ambient temperature for samples with the highest values for the hoop tensile strength (GFRC series 6 and 8).

The force-displacement behaviors of the specimen series 1 and 2 are similar, namely, there is linear behavior up to the cracking of the samples. In the cases of the specimen series 6 and 8, there is linear behavior up to the cracking of some layers of fibers but the samples are still not destroyed. With the continuing of the force, there is a bigger displacement and the curves continue to have a view which is similar to linear up to cracking of the samples. The failure of these composite split-disk samples was accompanied by a cracking noise and the appearance of a series of splits around the circumference of the pipe. This behavior is a result of the direction of the fibers, which in this case is 90°. The test results indicated an effect of the winding angle on the mechanical properties of composite specimens, namely, the bigger winding angle led to higher hoop tensile properties of filament-wound samples. In a polymer composite, each ply has a contribution to the whole strength, and when one of the layers in the structure starts to fail, it cracks the matrix around and there appears to be an increase in the strain. The strain response of the tube is restored but the load-carrying thickness of the tube is decreased due to the failure of one of the layers. As the wall thickness of the tube is decreased, it cannot carry more load anymore, and fails [6,7].

The transverse compressive strength of the specimens was calculated by using the following equation:
(5)
$$\sigma =\frac{F\max}{A}$$

where

*A* is the cross-sectional area,

(6)
$$A=\frac{\pi }{4}\left(OD^{2}-ID^{2}\right)$$

and *ID* and *OD* are the average inner and outer diameters, respectively.

For the determination of transverse compressive properties of specimens, filament-wound cylinders of 100 mm in diameter and 140 mm in length bonded into two end fixtures were tested from each testing group. The transverse compressive strength was determined from the maximum load carried before failure. Mainly, the ultimate compressive strength of the specimens was determined. In addition, the average of these results was calculated for each group, and with the aid of these data the general behavior of the specimens was determined. The overall results for the compressive properties of composite pipes with an experimental matrix are presented in Table 8.

By analyzing the obtained regression equation by implementing the 2^3^ full factorial experimental design, in coded variables, *y_ck_* is

(7)
$$y_{ck}=189.1138-8.1962x_{1}+11.1238x_{2}-95.7875x_{3}+11.2112x_{1}x_{2}+8.9375x_{1}x_{3}$$

and in engineering or natural variables, *y_tn_* is

(8)
$$y_{cn}=368.4662-7.6065x_{1}-0.5817x_{2}-2.7671x_{3}+0.1095x_{1}x_{2}+0.0284x_{1}x_{3}$$


It can be found out that the main contribution to the compression strength (σ) is given by the process parameter *x_3_*, i.e., the winding angle of the of the fibers. The transverse compression strength is inversely proportional to the winding angle of the fibers, which means that the higher the winding angle of the fibers is, the lower is the compression strength of the composite pipes. The influence of the other parameters *x_1_* and *x_2_* is the lowest. From the results shown in Table 4 it can be noticed that specimens 5–8 (wound with a 10° angle) had shown the highest values for compressive strength which correspond with the factor influences from regression equation. On contrary, specimens 1–4 wound with a 90° angle had shown a much lower value by almost 50%. From the received results it can be noticed that transverse compression properties of composite specimens significantly depended on winding angles in filament winding technology. Additionally, it can be noticed that there is a slight influence of the fiber tension and winding velocity of the fibers on the compression strength. However, the interaction of the two factors has a negligible positive effect on the response. Composite pipe is assumed to be a linear elastic material and its mechanical parameters can be considered to be insensitive to the stress state. In general condition, GFRC pipe can be seen as a linear elastic material in the fiber direction, but under uniaxial stresses it can have a nonlinear performance [12,24]. That is because, in the two phases of materials that compose GFRC pipes, the mechanical parameters of the fiber cannot be affected by the stress state, yet those of the resin are sensitive to the stress state. When the GFRC pipe is axially loaded, the variation of the stress state causes the variation of the resin mechanical parameters and, accordingly, the variation of the GFRC mechanical parameters [27,28,29].

Figure 14 shows a typical compressive stress–strain diagram for samples with higher values for the compressive strength (GFRC series 3 and 4) and for samples with lower values for the compressive strength (GFRC series 5 and 6).

As it can be seen from Figure 14, the failure stress is significantly higher for the composite tubular samples wound with an angle of 10° compared to specimen series 5 and 6 wound with an angle of 90°. The stress–strain behavior of the specimens is similar, namely, there is a linear behavior up to the cracking of the first reinforcing strand. During the filament winding process, as the tube diameter increases, the final layers are wound on a larger circumference; thus, the distance between adjacent strands in the layer increases. The empty distances between strands form suitable places for resin accumulation [13]. Therefore, for the specimen series 5 and 6, the fiber breakage was very limited, and the specimens failed due to the rupture of the resin-rich regions, which causes the formation of less macroscopic damage on the specimens, and low failure strength. For the tubular samples series 3 and 4 with the continuing of the force in the direction of the reinforcing strands which carry the compressive force, the curves have an appearance like a zigzag, which corresponds to the cracking of the other fiber layers. The failure of these tubular samples was accompanied by a cracking noise. In the filament winding process, the reinforcing strands are laid down on the mandrel without any crimp since the winding is carried out in one direction and strands do not interlace each other. Every layer consists of sets of strands which are not intertwined together. This inherent feature makes the filament-wound composites less resistant to delamination and crack propagation, since they lack interlaced points which can act as stop points in the crack path [14,15,16,17,18]. Therefore, in the filament-wound structure wound with an angle of 10°, the compressive force has a more destructive effect.

To validate the implementation of the FFED in the study and the assumed model, the theoretically calculated results are compared with experimental values for composite pipes with winding angle of the fiber of 10, 25, 45, 75, and 90°, manufactured with a winding velocity of 5.25 m/min and a fiber tension of 34 N. This comparison can be done with any other value for the fiber tension or winding velocity as long as it is within the study domain. The results are presented in Figure 15.

As can be seen from Figure 15, there is a good agreement between calculated and experimental values. All calculated values are placed in a straight line, which is in accordance with the assumed model of the experiment and are in close proximity of the experimental data. Based on the obtained regression equations in engineering or natural variables, we could do the design of composites. Therefore, for a given request for hoop tensile or compressive strengths, by substitution of *y_tn_* and *y_cn_* in the Equations (3) and (6), the winding angle of the fibers of the composite pipes can be calculated and then the appropriate winding angle will be used in the fabrication of the composite pipes. In addition, *y_tn_* and *y_cn_*, i.e., values of hoop tensile or compressive strengths, can be calculated for a given winding angle (*x_3_* factor). In both the above cases, the winding velocity and fiber tension has to be fixed at constant values (5.25 m/min and 34 N, respectively), for the most favorable outcome.

According to all obtained results in this investigation for mechanical and thermal characteristics, it can be concluded that from the mechanical point of view there are significant differences in the filament-wound pipes with different fiber orientations. For the tensile properties, the filament-wound pipes wound with an angle of 90° had shown much better values than pipes wound with an angle of 10°. From the received results, it can be noticed that tensile properties of composite specimens depended on winding angles in filament winding technology, namely, the bigger winding angles lead to higher hoop tensile properties of filament-wound tubular samples. In addition, it can be noticed that the fiber tension and velocity of the filament winding do not influence the tensile strength of the specimens. For the compression properties, it can be noticed that transverse compression properties of composite specimens depended on winding angles in filament winding technology, namely, the lower winding angles lead to higher transverse compression properties of filament-wound tubular samples. In addition, it can be noticed that there is a slight influence of fiber tension on compression strength. That means that higher fiber tension leads to higher value of the compression strength. The velocity of the filament winding does not influence the compressive strength of the specimens. From the thermal characterization, it can be concluded that the all filament-wound pipes have good thermal stability, and their complete weight losses were observed at temperature intervals from 600 °C to 1000 °C. The values for the glass transition temperature (Tg) of the analyzed composites are similar for all configurations of filament-wound pipes, which indicate that the degree of crosslinking is already reached in all composites. The values of the heat capacity of the composites prove that a high degree of crosslinking has already been reached in all composites. The glass transition temperature (Tg) in the composites increases with the increasing of the heating rate. The crosslinking reaction between the resin and fibers in the filament-wound pipes produced with lower and upper level of velocity of winding results in minor changes on the glass transition temperature. Obviously, the velocity of winding does not influence the glass transition characteristics.

## 4. Conclusions

Experimental measurements of the mechanical properties of composite pipes for determined ranges of winding parameters have been carried out implementing 2^3^ full factorial experimental designs. Regression equations were established for hoop tensile and transverse compression strengths as a function of the winding velocity, fiber tension and winding angle of the fibers. The experimental procedure described in the present work is sufficient to show the influence of the winding parameters on the tensile and compression properties of composite pipes produced by filament winding technique.

From this study the following conclusions are drawn:(1)From the thermal characterization, it can be concluded that all the produced filament-wound pipes have good thermal stability, and their complete weight loss was observed at temperature intervals from 600 °C to 1000 °C.(2)By using TGA, we have received 20–25 weight percentage of glass fibers which we have expected during the production procedure.(3)The values for the glass transition temperature (Tg) of the analyzed composites are similar for all configurations of filament-wound pipes, which indicate that the degree of crosslinking has already been reached in all composites.(4)The values of the heat capacity of the composites prove that a high degree of crosslinking has already been reached in all composites and there are not volatile materials.(5)The glass transition temperature (Tg) in the composites increases with the increasing of the heating rate.(6)The tensile and compression test results indicated that the change of the winding angle causes a huge variation in the final mechanical results, whereas the influence of the other two parameters—winding velocity and fiber tension—is much lower, and the interaction of the factors has a negligible effect on the response.(7)It was observed that if the study domain is precisely established (narrow enough), the full factorial experimental design can be employed to give good approximation of the response, i.e., stress of peak values.(8)Filament-wound composite pipes represent a good potential for utilization as loaded elements in the construction industry.(9)Based on the mechanical as well as thermal properties of these composite pipes, it can be concluded that they can be used for transportation of hot fluid under pressure.

## Figures and Tables

**Figure 1 polymers-15-02829-f001:**
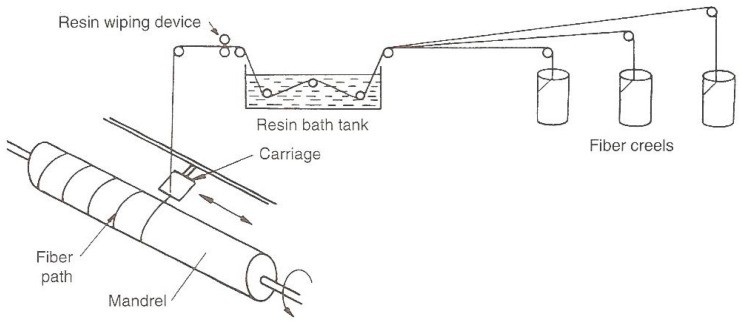
Schematic presentation of the filament winding technology.

**Figure 2 polymers-15-02829-f002:**
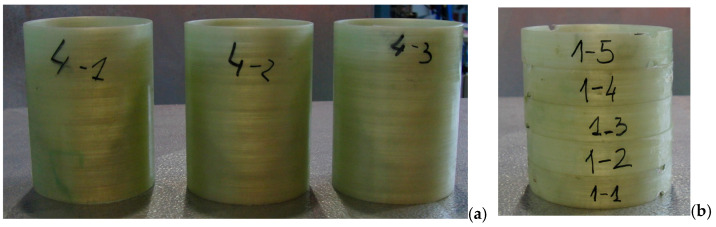
Prepared specimens: (**a**) pipes and (**b**) NOL rings.

**Figure 3 polymers-15-02829-f003:**
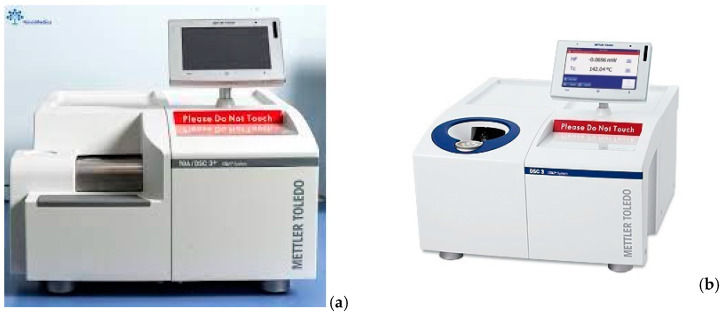
TGA (**a**) and DSC (**b**) instruments.

**Figure 4 polymers-15-02829-f004:**
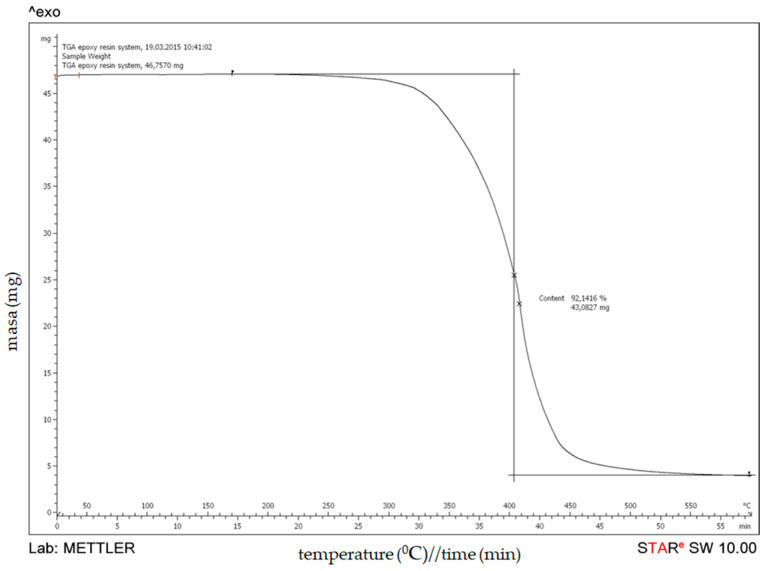
Thermogravimetric curve of an uncured epoxy resin system: weight loss (mg) versus temperature.

**Figure 5 polymers-15-02829-f005:**
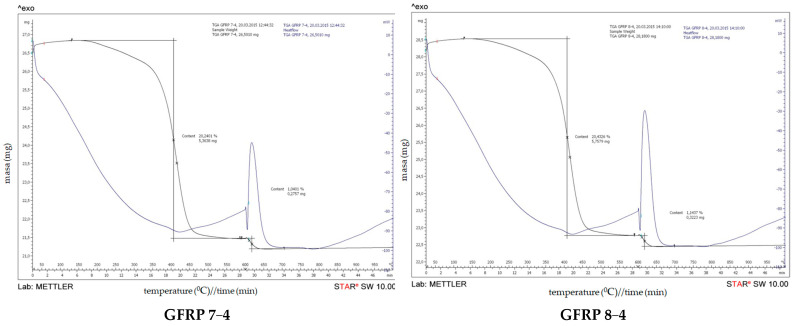
Thermogravimetric curve on filament-wound pipes: weight loss (mg) versus temperature.

**Figure 6 polymers-15-02829-f006:**
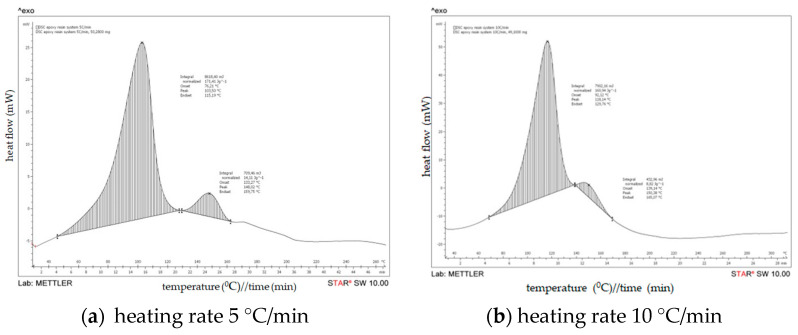
DSC results on uncured epoxy resin system.

**Figure 7 polymers-15-02829-f007:**
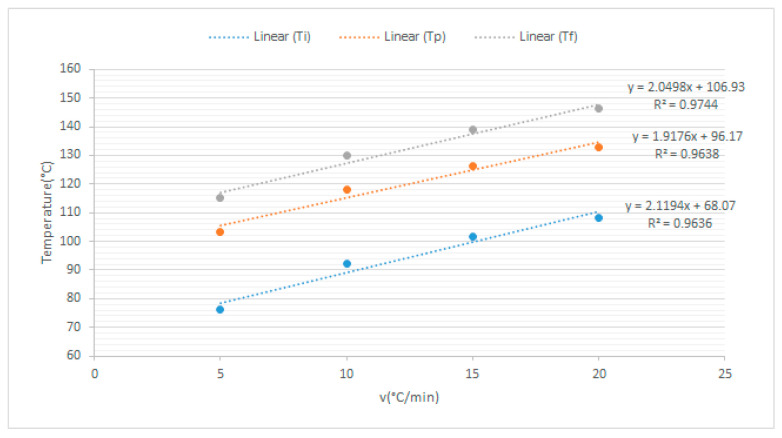
Temperature extrapolation curves at different heating rates.

**Figure 8 polymers-15-02829-f008:**
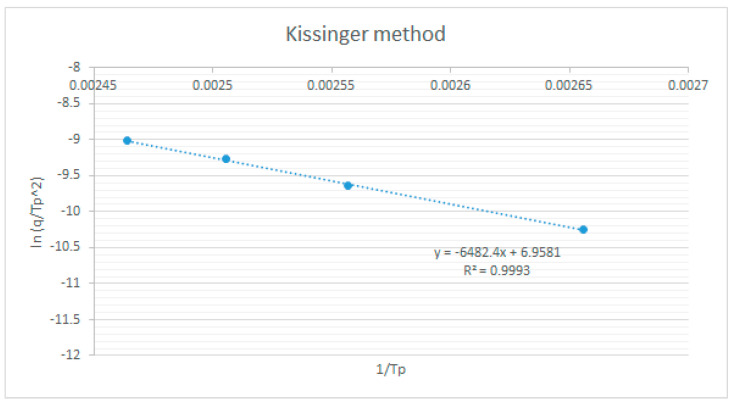
The curve of activation energy of the curing reaction was determined by the Kissinger method.

**Figure 9 polymers-15-02829-f009:**
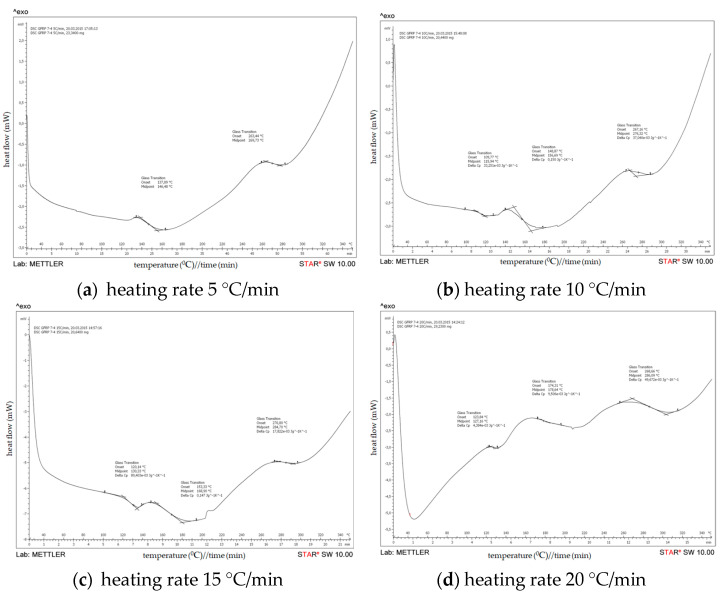
DSC results on filament-wound pipe—GFRP 7−4.

**Figure 10 polymers-15-02829-f010:**
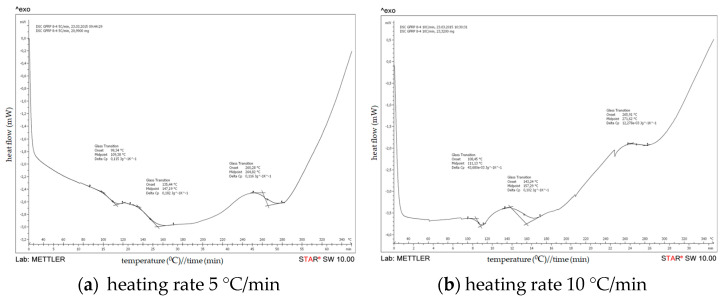
DSC results on filament-wound pipe—GFRP 8−4.

**Figure 11 polymers-15-02829-f011:**
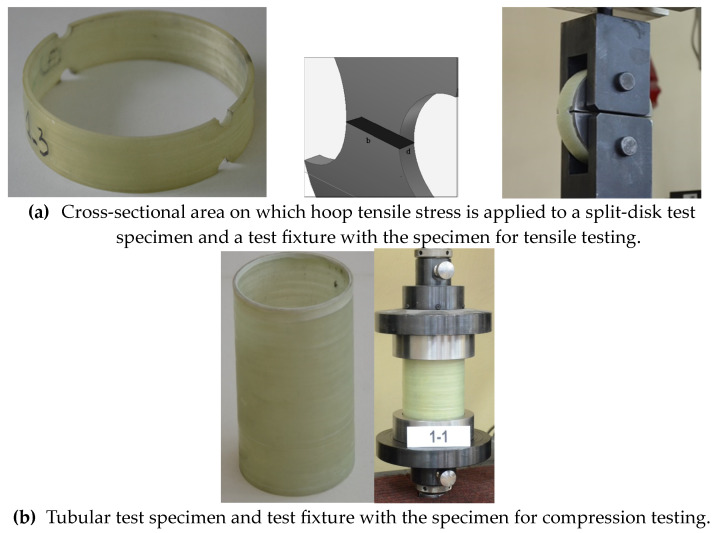
Illustration of the assembled tensile (**a**) and compression test fixture and specimens (**b**).

**Figure 12 polymers-15-02829-f012:**
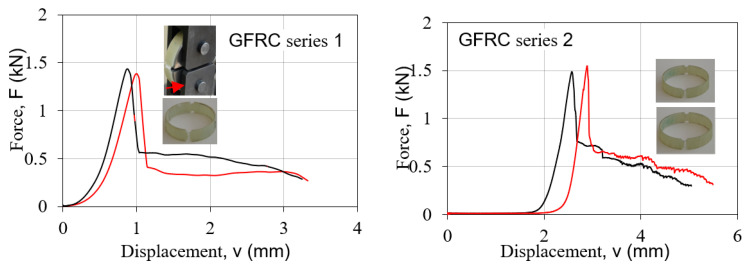
Force-displacement graphs of split-disk GFRC samples of series 1 and 2.

**Figure 13 polymers-15-02829-f013:**
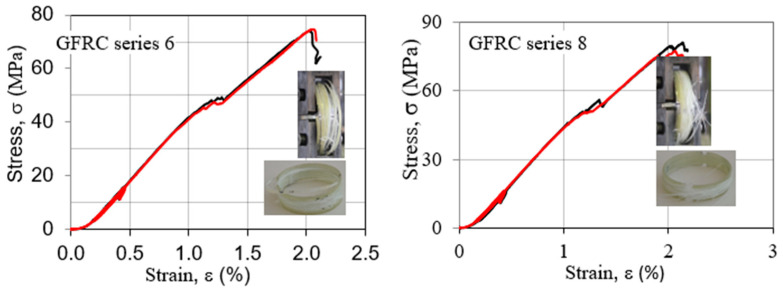
Tensile stress–strain curves of split-disk GFRC samples of series 6 and 8.

**Figure 14 polymers-15-02829-f014:**
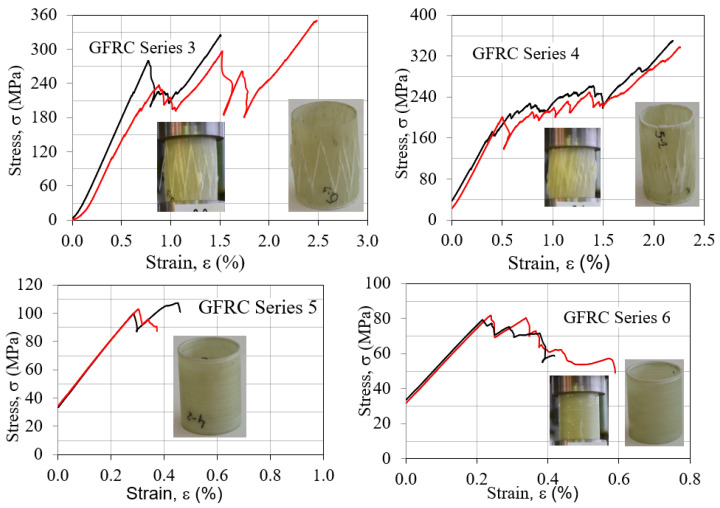
Compressive stress–strain curves of tubular samples of series 3, 4 and series 5, 6.

**Figure 15 polymers-15-02829-f015:**
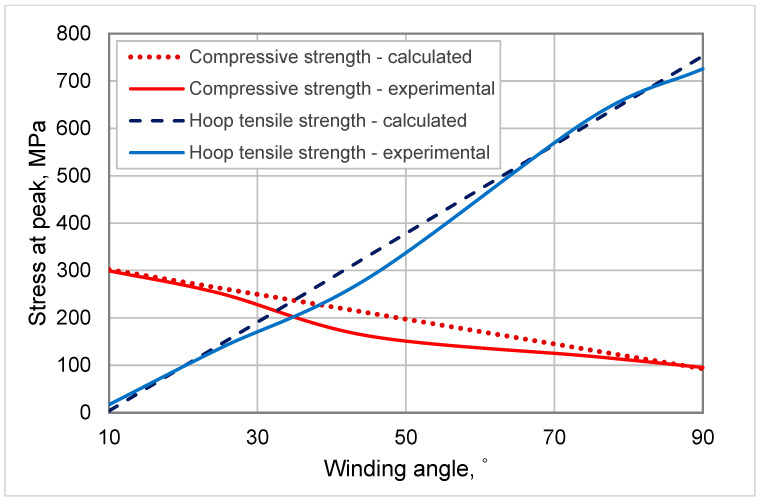
Mechanical properties vs. winding angle of the fiber in the composite pipes.

**Table 1 polymers-15-02829-t001:** Physicochemical properties of epoxy resin.

Density (g/cm^3^)	Viscosity (mPa·s)	Gel Time (min)	Pot Life (h)
1.15	600–1000 at 25 °C	50–60 at 80 °C	18–24 at 25 °C

**Table 2 polymers-15-02829-t002:** The main parameters of E-glass fiber.

Tensile Strength (MPa)	Linear Density (g/km)	Volume Density (g/cm^3^)	Elongation (%)
3100–3800	1200	2.6	4.5–4.9

**Table 3 polymers-15-02829-t003:** Coding convention of variables.

	Winding Speed, m/min	Fiber Tension, N	Winding Angle, °
Zero level, *x_i_* = 0	13.125	47	50
Interval of variation	7.875	13	40
High level, *x_i_* = +1	21	60	90
Lower level, *x_i_* = −1	5.25	34	10
Code	*x_1_*	*x_2_*	*x_3_*

**Table 4 polymers-15-02829-t004:** Thermal stability of filament-wound pipes based on glass fibers and epoxy resin.

Temperature Range (°C)	Weight Loss (%)		
	Epoxy	GFRP 7-4	GFRP 8-4
150–600	92.1416	20.2401	20.4326
600–1000	7.8584	1.0401	1.1437

**Table 5 polymers-15-02829-t005:** DSC analysis data obtained by heating an uncured epoxy resin system at a rate of 5, 10, 15 and 20 °C/min.

Specific Values and Extrapolation Results of Ti, Tp and Tf at Different Heating Rates				
Heating Rate q(°C/min)	Onset Ti (°C)	Peak Tp (°C)	Endset Tf (°C)	∆Hcure (Jg^−1^)
0	68.1	96.2	106.9	
5	76.21 (133.27)	103.50 (148.02)	115.19 (159.75)	171.41 (14.11)
10	92.12 (139.14)	118.14 (150.38)	129.76 (169.07)	160.94 (8.82)
15	101.52	126.12	138.98	260.84
20	108.40	132.80	146.28	237.96

**Table 6 polymers-15-02829-t006:** DSC analysis data obtained for GFRP 7-4 and 8-4 by heating rates of 5, 10, 15, and 20 °C/min.

Glass Transition	5 °C/min	10 °C/min	15 °C/min	20 °C/min	
Onset (°C)	135.4	143.2	159.8	167.1	GFRP 8-4
Midpoint (°C)	147.2	157.3	168.9	178.3	
∆Cp (Jg^−1^K^−1^)	0.182	0.102	0.035	0.060	
Onset (°C)	137.89	148.87	153.33	174.31	GFRP 7-4
Midpoint (°C)	146.48	156.69	169.90	178.64	
∆Cp (Jg^−1^K^−1^)		0.15	0.147	0.095	

**Table 7 polymers-15-02829-t007:** Experimental matrix with hoop tensile strength results of split-disk tests.

	*x_1_*	*x_2_*	*x_3_*	*x_1_ x_2_*	*x_1_ x_3_*	*x_2_ x_3_*	*x_1_ x_2_ x_3_*	σ Average (MPa)
1	−1	−1	−1	+1	+1	+1	−1	16.95
2	+1	−1	−1	−1	−1	+1	+1	17.00
3	−1	+1	−1	−1	+1	−1	+1	22.28
4	+1	+1	−1	+1	−1	−1	−1	25.32
5	−1	−1	+1	+1	−1	−1	+1	725.85
6	+1	−1	+1	−1	+1	−1	−1	854.60
7	−1	+1	+1	−1	−1	+1	−1	811.22
8	+1	+1	+1	+1	+1	+1	+1	881.20

**Table 8 polymers-15-02829-t008:** Experimental matrix with transverse compressive strength results of tubular tests.

	*x_1_*	*x_2_*	*x_3_*	*x_1_ x_2_*	*x_1_ x_3_*	*x_2_ x_3_*	*x_1_ x_2_ x_3_*	σ Average (MPa)
1	−1	−1	−1	+1	+1	+1	−1	299.40
2	+1	−1	−1	−1	−1	+1	+1	238.24
3	−1	+1	−1	−1	+1	−1	+1	304.68
4	+1	+1	−1	+1	−1	−1	−1	297.30
5	−1	−1	+1	+1	−1	−1	+1	95.40
6	+1	−1	+1	−1	+1	−1	−1	78.93
7	−1	+1	+1	−1	−1	+1	−1	89.77
8	+1	+1	+1	+1	+1	+1	+1	109.21

## Data Availability

Not applicable.

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
