# Peer review of "Effect of Process Parameters on Thermal and Mechanical Properties of Filament Wound Polymer-Based Composite Pipes"

_polymers, 2023, doi:10.3390/polym15132829_

Round 1

Reviewer 1 Report

This manuscript investigated the thermal and mechanical properties of filament wound polymer composites with three parameters. The results will benefit the industrial application. However, the number of experiments is not enough to support sound conclusions. Besides, there are some other issues that need to be addressed before publication:

1. Line 19. In the abstract, it is suggested to use no abbreviation without explaination. So, Tg should be replaced with glass transition temperature. 

2. The number of references in the Introduction part is low. Though it looks like there are more than 20 references, but before line 90, there are so much background information, but only four references. It is recommended to provide more references in the introduction part.

3. Line 130, is that 5.25 m/min? rather than 5,25 m/min? The coma made me confusing. The same issue occured in table 3.

4. Line 131, the symbol of degree (10 degree, 90 degree) should be used with the right one, instead of the number "0" 

5. Is there any reason for selecting the test values of the controlled parameters?

6. In all the figures, the title of the x-axis and y-axis should be provided. Also, the font of the labels in all the figures should be made large enough for the readers. As examples, Figure 4, figure 5, figure 7, etc. It is hardly to see clearly the labels of x and y-axis, and the title of the x- and y-axis are missing. 

7. Eq. (3-4), (7-8) are not linear models, there is an item of second order existing. But the authors claimed that a first order linear model is selected for this manuscript.

8. It is recommended to provide equations (3-4, 7-8)with actual items instead of the coding numbers so that it shows the meaning of the equation in practice. 

9. Line 455, is there a typo? Figure 6, or Figure 15? 

Author Response

Thank you for your review and your comments, I am sending you answers to your comments in an attached document and a corrected version of the paper.

Reviewer 2 Report

The investigation of the composite pipes mechanical properties and technology of it formation are considered in the paper « Effect of process parameters on thermal and mechanical properties of polymer-based composites in filament winding technology». The relevance of this paper is questionable, because of such investigation was carried out earlier by other authors. There are some questions, some of them listed below:

1. Authors should rewrite the introduction of the paper:

- It should be logic in the materials of the Introduction part. For example, how does the first paragraph relate to the second?

- Sometimes metal products are lighter than polymer composites, with the same strength (line 51-54).

- The authors should show the problem that has not yet been solved. The literature review of existing methods for solving the problem should be conduct more thorough. Only the concise description of the composite pipes formation technology and testing methods are considered in the Introduction part of that paper. Where is the problem?

- The filament winding technology formation of composite pipes has been used for a long time. So, such experiments have already been carried out. For what purpose authors are needed to repeat it? What problem do the authors want to solve?

2. Why is such a range of values chosen? (line 130-131).

The linear model choosing have to be substantiate (lines 131-134).  How does this choice relate to the range limits (lines 130-131)?

3. Please, show the calculation of the required tests number (line 135).

4. Only the one point in the range of parameters is choosing (lines 140-142). It looks unreasonable.

5. Why the thermal analysis method is used in that paper? What question did the authors seek an answer to using it?

6. Why did the authors use the TGA method? The samples drying was carried out at 140 degrees temperature.

7. How did the results obtained in section 3.1.1 a (lines 187-211) help the authors answer to the purpose of the paper?

8. Why do research when you know the answer? (lines 351-353).

Author Response

(The authors gave the same response as above.)

Round 2

Reviewer 1 Report

All the comments are solved. 

Author Response

Dear, 
All the comments are resolved.  
Thank you for review and suggest. 
Thank in advance  

Reviewer 2 Report

Unfortunately, the authors of the paper did not answer correctly all the questions. For example, the reviewer is not satisfied with the answers to his questions:
- point 3,
- point 4 (the calculation is incorrect, you need to use the theory of combinatorics),
- point 6, 7 and 8.

Author Response

Dear, 

Thank you for review and suggest. 
We have answered your questions and included them in the introduction and conclusion for clarity.
Hopefully it's clearer now

Thank in advance  

Round 3

Reviewer 2 Report

The authors have corrected the paper well. Now the paper can be published